# Unsupervised Lexical Simplification with Context Augmentation

**Takashi Wada**[1,2]     **Timothy Baldwin**[1,2]     **Jey Han Lau**[1]

[1] School of Computing and Information Systems, The University of Melbourne
[2] Department of Natural Language Processing, MBZUAI

twada@student.unimelb.edu.au     tb@ldwin.net     jeyhan.lau@gmail.com

## Abstract

We propose a new unsupervised lexical simplification method that uses only monolingual data and pre-trained language models. Given a target word and its context, our method generates substitutes based on the target context and also additional contexts sampled from monolingual data. We conduct experiments in English, Portuguese, and Spanish on the TSAR-2022 shared task, and show that our model substantially outperforms other unsupervised systems across all languages. We also establish a new state-of-the-art by ensembling our model with GPT-3.5. Lastly, we evaluate our model on the SWORDS lexical substitution data set, achieving a state-of-the-art result.[1]

## 1 Introduction

Lexical simplification is the task of replacing a word in context with an easier term without changing its core meaning, to make text easier to read for non-technical audiences, non-native speakers, or people with cognitive disabilities (e.g. dyslexia).

One common approach (Li et al., 2022; Qiang et al., 2020a,b) is to use a masked language model (MLM) such as BERT (Devlin et al., 2019) and predict substitutes via word prediction over the masked target word. However, one limitation is that it critically relies on the target context being discriminative of the semantics of the target word, which is not always the case. Given this, we propose a new unsupervised method that performs context augmentation. Specifically, we sample sentences that contain the target word from monolingual data, and identify substitutes that can replace the word in the target and sampled sentences. Based on our experiments in English, Portuguese, and Spanish over the TSAR-2022 shared task (Saggion et al., 2022), we show that our model comfortably outperforms other unsupervised models. We also es-

tablish a new state-of-the-art by ensembling our model with InstructGPT (Ouyang et al., 2022), and further demonstrate the effectiveness of the method over the related task of lexical substitution.

## 2 Method

We propose a *fully* unsupervised model using pre-trained language models (without fine-tuning) and monolingual data. Given the target word $x$ and context $c_x$, our model generates substitutes for $x$ based on not only $c_x$ but also augmented contexts sampled from monolingual data.

### 2.1 Generation Based on the Target Context

To generate substitutes of $x$ given $c_x$, we extend the lexical substitution approach of Wada et al. (2022),[2] which generates an aptness score $S(y|x, c_x)$ for each word $y \in V$ as follows:[3]

$$S(y|x, c_t) = \max_k \cos(f^k(y), f(x, c_x)),\,[4] \quad (1)$$

where $\cos$ denotes cosine similarity; $f(x, c_x)$ denotes the contextualised embedding of $x$ in $c_x$;[5] and $f^k(y)$ denotes the *decontextualised* embeddings of $y$, represented by $K$-clustered embeddings: $f^1(y), ... f^K(y)$, which are obtained by first sampling 300 sentences that contain $y$ from monolingual corpora, and clustering the contextualised embeddings of $y$ using $K$-means ($K = 4$). For each cluster $k$, $f^k(y)$ is calculated as $\frac{1}{|C_{y,k}|} \sum_{c'_y \in C_{y,k}} f(y, c'_y)$, where $C_{y,k}$ denotes sentences that contain $y$ and belong to the cluster $k$.

While this is the state-of-the-art unsupervised method on the SWORDS lexical substitution data

---

[2]Lexical substitution is closely related to lexical simplification, with no constraint on the lexical complexity of $x$.

[3]We set $|V|$ to 20,000 and 30,000 for the lexical simplification and substitution tasks, respectively.

[4]We also add the *global similarity* term to Eqn. (1) as proposed by Wada et al. (2022).

[5]Obtained by feeding $c_x$ into a pre-trained LM $f$ and averaging the embeddings of $x$ across multiple layers.

set (Lee et al., 2021), one major limitation is that $S(y|x, c_x)$ in Eqn. (1) heavily depends on $f(x, c_x)$, suggesting that if the meaning of $x$ is not well captured by $f(x, c_x)$, it may retrieve erroneous substitutes. In fact, this is often the case in lexical *simplification*, where $x$ is usually a rare word and gets segmented into subword tokens (in which case $f(x, c_x)$ is represented by the average of the subword embeddings). For instance, given the target word *bole*, the model retrieves *toe* as one of the top-10 substitutes, likely because the segmented *bol ##e* and *to ##e* share the same subword *##e*, suggesting that words that share the same token(s) tend to have similar representations regardless of their semantic similarity. To mitigate this, when $x$ is tokenised into multiple tokens, we add the term $\alpha\cos(E(x), E(y))$ to Eqn. (1), where $\alpha$ is a scalar value and $E(x)$ and $E(y)$ are pre-trained static embeddings of $x$ and $y$; we use fastText (Bojanowski et al., 2017) for this purpose. Since static embeddings are pre-trained with a large vocabulary size (e.g. 200k words), they tend to represent the semantics of rare words better than averaging embeddings of their (suboptimally tokenised) subwords.[6]

We tune $\alpha$ on the dev set and set it to 0.2, 0.7 and 0.6 for English, Spanish, and Portuguese, respectively. For embedding model $f$, we use DeBERTa-V3 (He et al., 2023) for English, and monolingual BERT models for Spanish and Portuguese (Cañete et al., 2020; Souza et al., 2020). We extract the $M_1 = 15$ words with the highest scores.

## 2.2 Generation with Context Augmentation

Following previous work on lexical simplification (Li et al., 2022; Qiang et al., 2020a,b), we also generate substitutes based on MLM prediction, by replacing $x$ with a mask token and performing word prediction. In this approach, the predictions are not affected by the embedding quality or tokenisation of $x$. However, if we rely solely on the target context $c_x$ as in previous work, the model has difficulty predicting substitutes when the context is not very specific; e.g. *The bole was cut into pieces*.[7] To address this problem, we perform context augmentation using monolingual data. Following the process of generating decontextualised embeddings in Wada et al. (2022), we sample 300 sentences that contain $x$ from monolingual corpora and cluster them using $K$-means ($K = 4$).[8] For each sentence in cluster $k$, we replace $x$ with a mask token and feed it into T5 (Raffel et al., 2020) in English, or mT5 (Xue et al., 2021) in Spanish and Portuguese, to generate 20 substitutes using beam search,[9] and retain those that contain only one word (which can comprise multiple subwords). Then, within each cluster $k$, we aggregate the substitutes across all sentences $c'_x \in C_{x,k}$ and extract the $M_2 = 25$ most-generated words. For each substitute candidate $y$, we calculate the score $\tilde{S}(y|x, c_x)$ as:

$$\tilde{S}(y|x, c_x) = \sum_k w_k \sum_{c'_x \in C_{x,k}} \mathrm{I}(y|c'_x)), \quad (2)$$

where $\mathrm{I}(y|c'_x))$ denotes a function that returns 1 if $y$ is generated by T5 given the context $c'_x$, and 0 otherwise; and $w_k$ denotes the number of substitutes in the cluster $k$ that overlap with the $M_1$ words generated from the target context $c_x$ in Section 2.1.[10] Here, $w_k$ roughly corresponds to the semantic relevance of the cluster $k$ to $c_x$; e.g. if $w_k = 0$, the candidates in the cluster $k$ would reflect a different sense of $x$ from the one in the target context and hence is not considered.[11] Intuitively, this scoring function favours substitutes that appear frequently in sampled contexts, weighted by how semantically relevant the substitute's cluster is to the original context — we will show its effectiveness with an example in Section 4. Finally, we retrieve the $M_2$ words with the highest scores and combine them with the $M_1$ candidates generated from $c_x$.

## 2.3 Reranking

Given $M_1 + M_2$ candidates[12] (potentially with overlap), we rerank them using four different metrics: (i) embedding similarity; (ii) LM perplexity; (iii)

---

[6] While fastText similarly makes use of character $n$-grams to represent words, it also trains a unique representation for each word, which is not shared with any other words (e.g. the word embedding of *her* is constructed by its character $n$-grams plus the special sequence *<her>*, where < and > correspond to the beginning and end of token). We also tried using GloVe (Pennington et al., 2014) instead of fastText in English, and observed comparable results.

[7] While previous work concatenates the masked sentence with the original (unmasked) sentence, it does not completely solve this problem.

[8] Note that Wada et al. (2022) sample sentences for generating $f^k(y)$, not for augmenting the contexts of $x$.

[9] In English, when the mask token directly follows the article *an* or *a*, we replace one of them with the other and feed the modified sentence to T5 and generate another 20 outputs. This way, we can mitigate the morphophonetic bias reported in Wada et al. (2022) (e.g. most of the generated substitutes for *accord* start with a vowel sound).

[10] In English, $\max_k w_k$ ranges from 0 to 12 (5.9 on average) with $M_1 = 15$ and $M_2 = 25$.

[11] When $w_k = 0$ for all clusters, we set $w_k$ to 1.

[12] Following Wada et al. (2022), we discard lexically-similar candidates, as measured by edit distance.

word frequency; and (iv) $\tilde{S}(y|x, c_x)$ in Eqn. (2). For the first metric, we use the reranking method proposed by Wada et al. (2022). For each candidate $y$, they replace $x$ in $c_x$ with $y$ and calculate the cosine similarity between the contextualised embeddings $f(x, c_x)$ and $f(y, c_x)$.[13] For the LM perplexity metric, we replace $x$ in $c_x$ with a mask token and calculate the probability of generating $y$ using T5; this score helps measure the syntactic fit of $y$ in $c_x$. The third metric corresponds to the frequency of $y$ in monolingual data,[14] which serves as a proxy for lexical simplicity. Finally, the last metric measures how often $y$ can substitute $x$ in the augmented contexts. Using each metric, we obtain four independent rankings $R_1, R_2, R_3, R_4$ and calculate their weighted sum: $r_1 R_1 + r_2 R_2 + r_3 R_3 + r_4 R_4$, which is then sorted in ascending order to produce the final ranking. We tune the weights $\{r_1, r_2, r_3, r_4\}$ based on the dev set for each language; $\{5, 1, 1, 1\}$, $\{3, 1, 0, 3\}$, and $\{3, 1, 0, 2\}$ for English, Spanish and Portuguese, respectively.

## 3 Experiments

### 3.1 Data and Evaluation

We experiment on the TSAR-2022 shared task on multilingual lexical simplification (Saggion et al., 2022; Štajner et al., 2022; Ferrés and Saggion, 2022; North et al., 2022b). We use its trial data as our dev set (about 10 instances per language) and evaluate models on the test set, which contains about 370 instances per language. Evaluation is according to four metrics: **Accuracy@1** = % of instances for which the top-1 substitute matches one of the gold candidates; **Accuracy@k@top1** = % of instances where one of the top-$k$ substitutes matches the top-1 gold label; **Potential@k** = % of instances where at least one of the top-$k$ substitutes is included in the gold candidates; and **MAP@k** = the mean average precision of the top-$k$ candidates.

### 3.2 Baselines

We compare our method against several systems submitted to the shared task. In all languages, **UniHD** (Aumiller and Gertz, 2022) is by far the best system across all metrics. It prompts GPT-3.5 (*text-davinci-002*, a.k.a. InstructGPT: Brown et al. (2020); Ouyang et al. (2022)) to provide ten easier alternatives for the target word $x$ given $c_x$, in two

variants: **zero-shot** and **ensemble**. The former generates substitutes based on the target word and context only, whereas the latter ensembles the predictions with six different prompts and temperatures; among them, four prompts include one or two question–answer pairs retrieved from the dev set to allow InstructGPT to perform in-context learning (as detailed in Table 8 in Appendix). While the ensemble model achieves the best results across all languages, it is not exactly comparable with the other systems as InstructGPT is *supervised* on various tasks with human feedback. As such, we also include the second-best systems (which differ for each language) as baselines, namely: **MANTIS** (Li et al., 2022), **GMU-WLV** (North et al., 2022a), and **PresiUniv** (Whistely et al., 2022). We also include the shared task baseline **LSBert** (Qiang et al., 2020a,b). All of these systems are based on pretrained MLMs like BERT (Devlin et al., 2019) and RoBERTa (Liu et al., 2019), and three of them also employ static word embeddings similarly to our model. Lastly, we also include Wada et al. (2022) with and without fastText in our baselines.

### 3.3 Results

Table 1 presents the results in English, Portuguese, and Spanish. The first five rows are based on InstructGPT: the first two show the zero-shot/ensemble performance reported in Aumiller and Gertz (2022), and the next two show the results when we replace *text-davinci-002* with *gpt-3.5-turbo*. The results show that *gpt-3.5-turbo* substantially outperforms *text-davinci-002*. The last row for InstructGPT shows the result when we prompt the model to provide simplified alternatives for $x$ *without the target context* (shown as **"w/o context"**), which indicates that the model performs very well even without access to the target context. This result demonstrates that the model has memorised lists of synonyms, and that most instances are not very context-dependent; we will return to discuss this in Appendix B.

The next five rows (marked "Unsupervised") show the performance of the unsupervised models, including ours. Our model clearly outperforms the other systems across all languages. In English, it even outperforms the zero-shot GPT-3.5-turbo in Potential@3 (94.1 vs. 92.8) despite the substantial differences between these models in terms of the model size (i.e. 435M and 800M parameters are used for DeBERTa-V3 and T5, respectively, and

[13]As in Section 2.1, we add $\alpha \cos(E(x), E(y))$ when $x$ is segmented into multiple tokens.

[14]We use the *wordfreq* Python library (Speer, 2022).

| Model | ACC@1 | | | ACC@3@Top1 | | | MAP@3 | | | Potential@3 | | |
|---|---|---|---|---|---|---|---|---|---|---|---|---|
| | en | pt | es | en | pt | es | en | pt | es | en | pt | es |
| InstructGPT (UniHD) | | | | | | | | | | | | |
| GPT-3.5-text-davinci-002-zero | 77.2 | 63.6 | 57.1 | 57.1 | 51.6 | 45.1 | 50.9 | 41.1 | 35.3 | 89.0 | 78.6 | 69.0 |
| GPT-3.5-text-davinci-002-ens | 81.0 | 77.0 | 65.2 | 68.6 | 62.3 | 57.9 | 58.3 | 50.1 | 42.8 | 96.2 | 91.7 | 82.1 |
| GPT-3.5-turbo-zero | 82.8 | 79.4 | 64.4 | 68.1 | 63.6 | 50.5 | 60.9 | 51.1 | 45.2 | 92.8 | 88.8 | 75.0 |
| GPT-3.5-turbo-ens | 87.4 | 85.8 | 76.4 | 71.8 | 73.5 | 62.2 | 65.5 | 58.7 | 55.9 | 97.3 | **97.3** | 89.1 |
| GPT-3.5-turbo-ens (w/o context) | 82.6 | 84.5 | 76.1 | 70.8 | 69.8 | 59.5 | 64.8 | 57.3 | 54.1 | 94.6 | 94.9 | 88.3 |
| Unsupervised | | | | | | | | | | | | |
| MANTIS/GMU-WLV/PresiUniv | 65.7 | 48.1 | 37.0 | 53.9 | 39.6 | 32.9 | 47.3 | 28.2 | 21.5 | 87.7 | 68.7 | 58.4 |
| LSBert | 59.8 | 32.6 | 28.8 | 53.1 | 28.6 | 18.2 | 40.8 | 19.0 | 18.7 | 82.3 | 49.5 | 49.5 |
| Wada et al. (2022) | 64.1 | 39.3 | 21.7 | 50.7 | 32.9 | 14.7 | 43.3 | 24.1 | 13.0 | 86.6 | 59.4 | 31.0 |
| Wada et al. (2022) + fastText | 64.6 | 51.9 | 32.3 | 51.2 | 42.8 | 25.0 | 43.7 | 31.0 | 19.5 | 86.9 | 70.9 | 49.2 |
| **OURS** | 79.9 | 61.5 | 47.8 | 63.5 | 52.7 | 37.0 | 57.5 | 38.0 | 30.0 | 94.1 | 83.2 | 71.5 |
| GPT-3.5-turbo-ens + WordFreq | **89.3** | 85.6 | **78.3** | 73.2 | 74.9 | 65.2 | 68.2 | 59.9 | 57.6 | 97.9 | **97.3** | 89.4 |
| + OURS | **89.3** | **86.4** | 77.7 | **75.1** | **76.7** | **66.8** | **69.9** | **61.1** | **59.1** | **98.7** | **97.3** | **89.9** |

Table 1: The results on lexical simplification. "-zero/ens" denote the zero-shot/ensemble models, and "w/o context" indicates the performance without access to the target context. The best scores among InstructGPT and unsupervised models are underlined, and the overall best scores are boldfaced.

175B parameters for GPT-3.5) and the language resources to use (i.e. our model employs monolingual data only while GPT-3.5 is instructed with human feedback). The strong performance in English is largely owing to the use of better LMs (DeBERTa-V3 and T5) compared to the ones used in Spanish and Portuguese (BERT and mT5), as evidenced by the substantial performance drop when we use BERT and mT5 for English.[15] The comparison of Wada et al. (2022) with and without fastText demonstrates the effectiveness of including static embeddings, especially in Portuguese and Spanish. This is because the vocabulary size of Portuguese/Spanish BERT is much smaller than that of DeBERTa-V3 (30/31k vs. 128k), and a large number of target words are segmented into subwords and embedded poorly. Lastly, we try ensembling: (1) the six rankings from GPT-3.5-turbo-ens; (2) the word frequency ranking (which we find boosts performance); and (3) the final ranking of OURS. The last two rows show the performance for (1) + (2) vs. (1) + (2) + (3). The ensemble of eight rankings including our method establishes a new state-of-the-art across all languages in most metrics, suggesting that our model is somewhat complementary to InstructGPT. In Appendix, we provide more detailed results (Table 7) and error analysis (Appendix B).

| Model | Lenient | | Strict | |
|---|---|---|---|---|
| | $F_a$ | $F_c$ | $F_a$ | $F_c$ |
| GPT-3-davinci | **34.6** | 49.0 | 22.7 | 36.3 |
| GPT-3.5-turbo-ens | 32.5 | 67.5 | 27.2 | 51.2 |
| Wada et al. (2022) | 33.6 | 65.8 | 24.5 | 39.9 |
| Qiang et al. (2023) | – | – | 24.9 | 40.1 |
| **OURS** | 33.3 | 66.2 | 25.3 | 41.8 |
| **OURS** + GPT-3.5-turbo-ens | 33.1 | **69.8** | **28.2** | **52.2** |

Table 2: The results on lexical substitution.

### 3.4 Experiment on Lexical Substitution

We also evaluate our model on the English lexical *substitution* task over the SWORDS data set (Lee et al., 2021). For lexical substitution, there is no restriction on lexical simplicity, so we drop the word frequency feature in reranking (i.e. set $r_3$ to 0).[16] Table 2 shows the results in the *lenient* and *strict* settings.[17] $F_a$ and $F_c$ denote the F1 scores given two different sets of gold labels $a$ and $c$, where $a \subset c$. In the strict setting, our model outperforms the best unsupervised model of Wada et al. (2022) and also the (non-LLM) state-of-the-art *semi-supervised* model of Qiang et al. (2023), which employs BLEURT (Sellam et al., 2020) and a sentence-paraphrasing model, both of which are

---

[15]The exact scores are 69.4, 58.7, 45.0, and 88.2 for ACC@1, ACC@3@Top1, MAP@3, and Potential@3, resp.

[16]We also double $M1/M2$ to 30/50 to make sure that our model provides top-50 words, following Wada et al. (2022).

[17]In the lenient setting, generated words are filtered out if their aptness scores are not annotated in SWORDS, whereas in the strict setting, all words are considered in the evaluation.

| Method | ACC@1 | ACC@k@Top1 | | | MAP@k | | | Potential@k | | |
|---|---|---|---|---|---|---|---|---|---|---|
| | | k=1 | k=2 | k=3 | k=3 | k=5 | k=10 | k=3 | k=5 | k=10 |
| Soft Retrieval | **63.1** | **32.4** | **45.0** | **51.1** | **41.8** | **30.9** | **18.8** | **82.9** | **89.2** | 93.3 |
| Hard Retrieval | 60.5 | 30.7 | 43.3 | 50.9 | 40.3 | 29.7 | 18.1 | 82.4 | 86.6 | 91.9 |
| No Clustering | 62.4 | 31.6 | 44.7 | 51.0 | 41.3 | 30.8 | **18.8** | 82.4 | 88.9 | **94.2** |

Table 3: The results on the lexical simplification task using different cluster-retrieval methods in Eqn. (2). The scores are averaged over English, Portuguese, and Spanish. "Soft Retrieval" indicates our original method proposed in Eqn. (2), and "Hard Retrieval" denotes when we set $w_k = 1$ for the closest cluster and $w_k = 0$ otherwise. The last row indicates when we set $w_k = 1$ for all clusters, which is equivalent to performing no clustering.

| | |
|---|---|
| Context ($x = elite$) | Syria is overwhelmingly Sunni, but President Bashar Assad and the ruling elite belong to the minatory Alawite sect. |
| Wada et al. (2022) | **establishment**, hierarchy, wealthy |
| Cluster1 ($w_k = 0$) | special, military, small |
| Cluster2 ($w_k = 5$) | **class**, political, **privileged** |
| Cluster3 ($w_k = 1$) | exclusive, international, prestigious |
| Cluster4 ($w_k = 0$) | top, professional, great |
| Soft Retrieval | **class**, **privileged**, political |
| No Clustering | top, professional, exclusive |
| **OURS** | **class**, **establishment**, leadership |

Table 4: Top-3 substitutes generated based on the target context (Wada et al. (2022)) and the augmented contexts (Section 2.2). The values for $w_k$ denote the weights for each cluster in Eqn. (2). "OURS" reranks the candidates of Wada et al. (2022) and "Soft Retrieval". The words included in the gold labels are boldfaced.

pre-trained on labelled data. Lastly, we also ensemble our model with the six outputs of GPT-3.5, and establish a new state-of-the-art.

## 4 Ablation Study

We perform ablation studies on the effect of clustering in Eqn. (2), and present the results in Table 3; the scores are averaged over English, Portuguese, and Spanish. "Soft Retrieval" indicates the performance when we take the weighted sum of the clusters as we propose in Eqn. (2); "Hard Retrieval" denotes when we set $w_k = 1$ for the closest cluster and $w_k = 0$ otherwise; and "No Clustering" denotes when we set $w_k = 1$ for all the clusters, which is equivalent to performing no clustering. The table shows that our proposed method performs the best overall, albeit with a small margin over "No Clustering". In fact, this is more or less expected since the majority of target words are used in their predominant senses (as evidenced by the strong performance of GPT-3.5 w/o context in Ta-

ble 1), in which case, retrieving the most-generated words across all sampled sentences would suffice to produce good substitutes.

Table 4 shows one example where clustering plays a crucial role (more predictions plus another example are shown in Table 5 in Appendix A). In this example, the target word *elite* is used as a noun meaning "a select group", but all clusters except for Cluster 2 produce the substitutes for *elite* in adjectival senses. Therefore, if we naïvely aggregate the words across all clusters ("No Clustering"), we end up retrieving adjectives such as *top* and *professional*, whereas our weighted-sum approach ("Soft Retrieval") successfully extracts good substitutes from the relevant clusters.

## 5 Related Work

Recent lexical simplification models are based on generating substitute candidates using MLM prediction and reranking, using features such as fast-Text embedding similarities and word frequency (Qiang et al., 2020a; Li et al., 2022). Some also use external tools or resources such as POS taggers or paraphrase databases (Qiang et al., 2020b; Whistely et al., 2022). However, Aumiller and Gertz (2022) show that GPT-3.5 substantially outperforms previous models on the TSAR-2022 shared task. Similar to this work, our prior work (Wada et al., 2023) samples sentences from monolingual corpora and use them to paraphrase multiword expressions with literal expressions (composed of 1 or 2 words).

## 6 Conclusion

We propose a new unsupervised lexical simplification method with context augmentation. We show that our model outperforms previous unsupervised methods, and by combining our model with InstructGPT, we achieve a new state-of-the-art for lexical simplification and substitution.

# 7 Limitations

One limitation of our model is that it performs context augmentation using monolingual data, which incurs additional time and computational cost. However, if we construct a comprehensive list of complex words $X$ and sample sentences containing $x \in X$ in advance, we can pre-compute the generation counts: $\sum_{c'_x \in C_{x,k}} \mathrm{I}(y|c'_x)$ in Eqn. (2) without considering the target context $c_x$ (which is required to calculate $w_k$ only). Therefore, we can still generate substitutes in an online manner during inference as long as the target word $x$ is included in $X$.

Compared to the InstructGPT baseline, our model critically relies on word embeddings and MLM prediction, both of which hinge on word co-occurrence statistics. This sometimes results in wrongly predicting antonyms of the target word as substitutes due to the similarity of their surrounding contexts (e.g. *famed* for *infamous*; more specific examples and error types are shown in Appendix B). On the other hand, InstructGPT benefits from supervision with human feedback and also makes use of lexical knowledge provided in various forms of texts during pre-training, including dictionaries, thesauri, and web discussions about meanings of words.[18] This is clearly one of the reasons why InstuctGPT substantially outperforms the other unsupervised systems, including ours; in fact, we find that it performs extremely well even *without access to the target context* (Table 1), motivating a call for including more context-sensitive instances in lexical substitution/simplification data sets; more discussions follow in Appendix B.

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

## A Impact of Clustering

Table 5 shows two examples where clustering plays a crucial role; the first instance was partially shown in Table 4 and discussed in Section 4. In the second instance, Soft Retrieval retrieves substitutes that are relevant to the meaning of the target word *extend* given this particular context. Without clustering, on the other hand, we get a mixed bag of words that represent different senses of *extend* (with more frequent senses ranked higher). In both cases, GPT-3.5 produces context-aware substitutes, although this is not always the case (as we discuss in the next section), and some of the candidates do not fit naturally in context (e.g. *ruling class* is predicted as the best substitute for *elite* used in *ruling elite*).

## B Error Analysis

Table 6 shows some examples of outputs from our model and GPT-3.5-turbo-ens on the lexical simplification and substitution tasks. In the first example, our model generates near-synonyms of the target word *infamous* with both negative and positive connotative meaning (e.g. *notorious* and *famed*, respectively), while GPT-3.5 generates negative-connotation words like *disgraceful* only. This is because our model heavily relies on word representations and assigns high scores to those words that appear in similar contexts. Similarly, our model incorrectly predicts *notoriously* as a substitute for *infamous* despite its ungrammaticality in this context, likely due to the similarity of their embeddings; we surmise that using a part-of-speech tagger would help alleviate this problem.[19] In the second instance, our model is overly affected by the target context and generates words that often appear in similar contexts to the target context but have different semantics from the target word *strategic*, e.g. *global* and *economic*. In comparison, GPT-3.5 generates more correct substitutes; however, some of them do not sound quite natural in this context (e.g. *calculated*). This is more evident in the third instance, where some of the "gold" substitutes are semantically marked when put into this context — the original phrase *in defiance of calls* means "opposition against calls", but *in resistance of calls* and *in rebellion of calls* (both predicted by GPT-3.5 and included in the gold labels) do not sound natural. These examples suggest that human annotators are sometimes oblivious to the context and consider substitutes largely based on the out-of-context similarities of the words,[20] which motivates a call for revisiting the exact goal of lexical simplification/substitution and its annotation schemes, e.g. whether the words should be annotated based on the *similarity of lexical semantics* or *acceptability in context*. The same concern is also raised by the strong performance of GPT-3.5 without access to the target context (Table 1).

In the last three examples, which are taken from the SWORDS lexical substitution data set, the sensitivity of our model to context works favourably and results in better substitutes than GPT-3.5, which, in those examples, generates substitutes without considering the context very much (in contrast to the examples in Table 6). For these instances, we also tried using the ChatGPT web interface (the free version, accessed in May 2023) and found that its outputs are highly stochastic even with the same prompt:[21] sometimes it returns substitutes that are quite similar to the ones generated by GPT-3.5-turbo, and other times it generates more context-aware and accurate substitutes (e.g. *business* for *service* and *probably/presumably* for *likely*). As such, further investigation is needed to see how carefully the model pays attention to the context (given different prompts), and how well it works for instances that require a profound understanding of the context.

---

[19] Another idea is to increase the weight for the LM perplexity in reranking, but it comes with a trade-off as it increases the unwanted bias of the target context.

[20] Similarly, in Spanish and Portuguese, annotators suggest both masculine and feminine nouns given a nominal target word. Wada et al. (2022) make a similar observation in Italian.

[21] We have also found that it often outputs an opening line such as "Here are ten alternative words for ...", while GPT-3.5-turbo (accessed via OpenAI API) usually returns a list of substitutes only.

| | | | |
|---|---|---|---|
| Context ($x$ = *elite*) | | Syria is overwhelmingly Sunni, but President Bashar Assad and the ruling elite belong to the minatory Alawite sect. | |
| Gold | | upper class, class, establishment, nobility, aristocracy, circle, elect, group, high society, noble, privileged, rich, select, society, superiors | |
| Wada et al. (2022) | | **establishment**, hierarchy, wealthy, bureaucracy, apparatus, leadership, ruling, affluent, clergy, mafia | |
| T5 | Cluster1 ($w_k = 0$) | special, military, small, specialized, american, professional, secret, infamous, heroic, undercover | |
| | Cluster2 ($w_k = 5$) | **class**, political, **privileged**, **rich**, majority, minority, party, **group**, wealthy, liberal | |
| | Cluster3 ($w_k = 1$) | exclusive, international, prestigious, new, small, professional, top, large, **select**, special | |
| | Cluster4 ($w_k = 0$) | top, professional, great, competitive, good, high, olympic, pro, collegiate, excellent | |
| | Soft Retrieval No Clustering | **class**, **privileged**, political, **rich**, majority, minority, wealthy, exclusive, party, **group** top, professional, exclusive, international, special, great, prestigious, small, new, **select** | |
| OURS GPT-3.5-turbo-ens | | **class**, **establishment**, leadership, **rich**, hierarchy, **privileged**, bureaucracy, apparatus, family, clergy ruling class, **high society**, **aristocracy**, **upper class**, exclusive, **nobility**, **privileged**, privileged few, **establishment**, top brass | |
| Context ($x$ = *extend*) | | I would wish to extend my thoughts and prayers to the family and friends of the victim at this terrible time. | |
| Gold | | expand, offer, give, send, continue, convey, dedicate, relay, reveal, share | |
| Wada et al. (2022) | | express, expanded, expanding, **offer**, render, impart, **convey**, exert, confer, grant | |
| T5 | Cluster1 ($w_k = 1$) | reach, go, stretch, run, apply, **continue**, spread, move, amount, span | |
| | Cluster2 ($w_k = 0$) | increase, prolong, lengthen, improve, shorten, reduce, **continue**, stretch, change, make | |
| | Cluster3 ($w_k = 2$) | enhance, provide, apply, improve, increase, bring, **offer**, broaden, use, **give** | |
| | Cluster4 ($w_k = 4$) | **offer**, **give**, **send**, express, say, **convey**, wish, add, make, provide | |
| | Soft Retrieval No Clustering | **offer**, **give**, **send**, provide, express, apply, enhance, make, add, bring increase, improve, stretch, prolong, **give**, enhance, **continue**, **offer**, apply, provide | |
| OURS GPT-3.5-turbo-ens | | **offer**, express, **send**, spread, stretch, **convey**, **give**, **share**, lend, extensions **offer**, **send**, express, **convey**, **share**, stretch, lengthen, **give**, present, **expand** | |

Table 5: Examples of top-10 substitutes generated based on the target context (Wada et al. (2022); Section 2.1) and the augmented contexts (Section 2.2). "OURS" denotes the substitutes obtained by reranking the candidates of Wada et al. (2022) and "Soft Retrieval". The values for $w_k$ denote the weights for each cluster in Eqn. (2), which correspond to the number of the shared words between the top-15 words from Wada et al. (2022) and the top-25 words from each cluster. The words included in the gold labels are boldfaced.

| | |
|---|---|
| Context (x = infamous) | He has denied charges of genocide, murder acts of terror and other crimes against humanity.The most infamous of the charges accuses Mladic of overseeing the massacre of 8 thousand Muslim boys and men in Srebrenica in Eastern Bosnia in 1995. |
| OURS | **notorious**, iconic, famed, renowned, legendary, **controversial**, heinous, notoriously, dreaded, notoriety |
| GPT-3.5 | **notorious**, **disreputable**, scandalous, shameful, ill-famed, infamously known, **disgraceful**, **dishonorable**, ignominious, detestable |
| Context (x = strategic) | The Taliban said it was in response to Obama's visit and to the strategic partnership deal he signed with Afghan President Hamid Karzai, a pact that sets out a long-term U.S. role after most foreign combat troops leave by the end of 2014. |
| OURS | security, **political**, national, international, global, military, economic, **tactical**, operational, policy |
| GPT-3.5 | **important**, **crucial**, essential, vital, significant, **planned**, key, critical, **tactical**, **calculated** |
| Context (x = defiance) | Taliban bombers attacked a heavily fortified guesthouse used by Westerners in Kabul on Wednesday, announcing the start of their annual in defiance of calls from visiting US President Barack Obama that the war was ending. |
| OURS | **disregard**, violation, contravention, contempt, **spite**, rejection, breach, defying, **opposition**, contradiction |
| GPT-3.5 | **resistance**, **rebellion**, **opposition**, disobedience, challenge, noncompliance, insubordination, **refusal**, nonconformity, dissent |
| Context (x = form) | "Perhaps." "I think his face is his form of vanity. It's the reverse of you with those ridiculous stomach muscles. |
| OURS | **type**, mode, version, method, manner, means, style, way, kind, breed |
| GPT-3.5 | shape, figure, structure, configuration, physique, appearance, build, outline, contour, profile |
| Context (x = service) | The National Association of Diaper Services, Philadelphia, says that since January it has gotten more than 672 inquiries from people interested in starting diaper services. Elisa Hollis launched a diaper service last year because State College, Pa., where she lives, didn't have one. Diaper shortages this summer limited growth at Stork Diaper Services, Springfield, Mass., where business is up 25% in |
| OURS | **business**, **company**, provider, program, center, operation, system, delivery, site, agency |
| GPT-3.5 | assistance, support, aid, help, provision, maintenance, care, repair, supply, benefit |
| Context (x = likely) | Upjohn further estimated that about 50% of the employees who leave for early retirement may be replaced. As a result, Upjohn will likely trim only about 275 to 350 of its more than 21,000 jobs world-wide. In composite trading on the New York Stock Exchange yesterday, Upjohn shares rose 87.5 cents to $38.875 apiece. |
| OURS | **probably**, undoubtedly, probable, possibly, surely, certainly, perhaps, **presumably**, potentially, likelihood |
| GPT-3.5 | probable, expected, anticipated, possible, plausible, foreseeable, credible, presumed, expect, anticipate |

Table 6: Examples of outputs from our model and GPT-3.5-turbo-ens. The first and last three instances are from lexical simplification and substitution data sets, respectively. The words included in the gold labels are boldfaced.

| | Model | ACC@1 | ACC@k@Top1 | | | MAP@k | | | Potential@k | | |
|---|---|---|---|---|---|---|---|---|---|---|---|
| | | | k=1 | k=2 | k=3 | k=3 | k=5 | k=10 | k=3 | k=5 | k=10 |
| en | GPT-3.5-text-davinci-002-zero | 77.2 | 42.6 | 53.4 | 57.1 | 50.9 | 36.5 | 20.9 | 89.0 | 93.0 | 94.4 |
| | GPT-3.5-text-davinci-002-ens | 81.0 | 42.9 | 61.1 | 68.6 | 58.3 | 44.9 | 28.1 | 96.2 | 98.1 | 99.5 |
| | GPT-3.5-turbo-zero | 82.8 | 52.3 | 63.5 | 68.1 | 60.9 | 46.7 | 28.0 | 92.8 | 94.4 | 95.2 |
| | GPT-3.5-turbo-zero (w/o context) | 83.9 | 46.6 | 60.9 | 67.6 | 62.4 | 46.5 | 28.3 | 92.8 | 95.4 | 97.3 |
| | GPT-3.5-turbo-ens | 87.4 | 54.7 | 65.7 | 71.8 | 65.5 | 52.5 | 33.3 | 97.3 | 99.2 | **100.0** |
| | GPT-3.5-turbo-ens (w/o context) | 82.6 | 46.1 | 63.8 | 70.8 | 64.8 | 50.0 | 31.9 | 94.6 | 97.6 | 98.9 |
| | MANTIS | 65.7 | 31.9 | 45.0 | 53.9 | 47.3 | 36.0 | 21.9 | 87.7 | 94.6 | 97.9 |
| | LSBert | 59.8 | 30.3 | 44.5 | 53.1 | 40.8 | 29.6 | 17.6 | 82.3 | 87.7 | 94.6 |
| | Wada et al. (2022) + fastText | 64.6 | 27.6 | 44.2 | 51.2 | 43.7 | 32.2 | 19.8 | 86.9 | 91.4 | 95.2 |
| | **OURS** (BERT + mT5) | 69.4 | 33.8 | 49.1 | 58.7 | 45.0 | 34.2 | 21.0 | 88.2 | 94.9 | 98.1 |
| | **OURS** (DeBERTa-V3 + T5) | 79.9 | 43.7 | 57.9 | 63.5 | 57.5 | 42.9 | 26.6 | 94.1 | 97.6 | 98.9 |
| | WordFreq + GPT-3.5-turbo-ens | **89.3** | 55.0 | **68.1** | 73.2 | 68.2 | 54.1 | 34.6 | 97.9 | **99.7** | 99.7 |
| | + **OURS** | **89.3** | **56.0** | **68.1** | **75.1** | **69.9** | **55.2** | **35.4** | **98.7** | **99.7** | **100.0** |
| pt | GPT-3.5-text-davinci-002-zero | 63.6 | 37.2 | 46.3 | 51.6 | 41.0 | 28.9 | 16.2 | 78.6 | 81.8 | 84.2 |
| | GPT-3.5-text-davinci-002-ens | 77.0 | 43.6 | 53.5 | 62.3 | 50.1 | 36.2 | 21.7 | 91.7 | 94.9 | 97.9 |
| | GPT-3.5-turbo-zero | 79.4 | 46.5 | 59.1 | 63.6 | 51.0 | 37.1 | 21.7 | 88.8 | 89.8 | 91.4 |
| | GPT-3.5-turbo-zero (w/o context) | 78.6 | 44.1 | 57.2 | 62.8 | 51.4 | 38.0 | 22.1 | 90.6 | 92.8 | 94.4 |
| | GPT-3.5-turbo-ens | 85.8 | 48.7 | 65.2 | 73.5 | 58.7 | 44.5 | 26.8 | **97.3** | 98.4 | 98.9 |
| | GPT-3.5-turbo-ens (w/o context) | 84.5 | **49.5** | 64.4 | 69.8 | 57.3 | 43.2 | 25.8 | 94.9 | 96.5 | 97.6 |
| | GMU-WLV | 48.1 | 25.4 | 37.2 | 39.6 | 28.2 | 19.7 | 11.5 | 68.7 | 75.7 | 84.0 |
| | LSBert | 32.6 | 15.8 | 23.3 | 28.6 | 19.0 | 13.1 | 7.8 | 49.5 | 58.0 | 67.4 |
| | Wada et al. (2022) + fastText | 51.9 | 27.0 | 37.7 | 42.8 | 31.0 | 22.5 | 12.9 | 70.9 | 77.3 | 85.3 |
| | **OURS** (BERT + mT5) | 61.5 | 31.6 | 45.2 | 52.7 | 38.0 | 28.3 | 16.9 | 83.2 | 89.3 | 92.8 |
| | WordFreq + GPT-3.5-turbo-ens | 85.6 | 48.4 | 65.5 | 74.9 | 59.9 | 45.4 | 27.4 | **97.3** | 98.1 | **99.2** |
| | + **OURS** | **86.4** | 49.2 | **66.8** | **76.7** | **61.1** | **46.8** | **28.1** | **97.3** | **98.7** | **99.2** |
| es | GPT-3.5-text-davinci-002-zero | 57.1 | 30.7 | 39.7 | 45.1 | 35.3 | 24.5 | 13.8 | 69.0 | 71.5 | 74.5 |
| | GPT-3.5-text-davinci-002-ens | 65.2 | 35.1 | 51.1 | 57.9 | 42.8 | 32.4 | 19.7 | 82.1 | 88.9 | 94.0 |
| | GPT-3.5-turbo-zero | 64.4 | 33.4 | 43.8 | 50.5 | 45.2 | 33.0 | 18.8 | 75.0 | 77.4 | 78.0 |
| | GPT-3.5-turbo-zero (w/o context) | 74.7 | 40.2 | 50.3 | 57.1 | 52.5 | 37.9 | 22.0 | 87.2 | 88.6 | 89.9 |
| | GPT-3.5-turbo-ens | 76.4 | 39.9 | 54.3 | 62.2 | 55.9 | 42.2 | 25.7 | 89.1 | 92.4 | 94.6 |
| | GPT-3.5-turbo-ens (w/o context) | 76.1 | **42.4** | 52.7 | 59.5 | 54.1 | 41.3 | 25.0 | 88.3 | 91.8 | 94.6 |
| | PresiUniv | 37.0 | 20.4 | 27.7 | 32.9 | 21.5 | 15.0 | 8.3 | 58.4 | 64.7 | 72.6 |
| | LSBert | 28.8 | 9.5 | 14.4 | 18.2 | 18.7 | 13.5 | 8.0 | 49.5 | 61.1 | 74.7 |
| | Wada et al. (2022) + fastText | 32.3 | 15.5 | 21.2 | 25.0 | 19.5 | 13.5 | 7.8 | 49.2 | 53.5 | 62.8 |
| | **OURS** (BERT + mT5) | 47.8 | 22.0 | 31.8 | 37.0 | 30.0 | 21.5 | 13.0 | 71.5 | 80.7 | 88.3 |
| | WordFreq + GPT-3.5-turbo-ens | **78.3** | 42.1 | 54.1 | 65.2 | 57.6 | 43.9 | 26.6 | 89.4 | 92.9 | **96.2** |
| | + **OURS** | 77.7 | 41.8 | **56.5** | **66.8** | **59.1** | **44.8** | **27.2** | **89.9** | **93.2** | **96.2** |

Table 7: The results on the TSAR-2022 lexical simplification task. "-zero/ens" denote the zero-shot/ensemble models proposed by UniHD (Aumiller and Gertz, 2022), and "w/o context" indicates the performance without access to the target context. The best scores in each language are boldfaced.

| GPT-3.5-turbo Prompt With Context |
|---|
| {"role": "system", "content": "You are a helpful assistant."}
{"role": "user", "content": "Context: A local witness said a separate group of attackers disguised in burqas — the head-to-toe robes worn by conservative Afghan women — then tried to storm the compound.\nQuestion: Given the above context, list ten alternative words for "disguised" that are easier to understand.\n"}
{"role": "assistant","content": "1. concealed\n2. dressed\n3. hidden\n4. camouflaged\n5. changed\n6. covered\n7. masked\n8. unrecognizable\n9 converted\n10. impersonated\n\n"}
{"role": "user", "content": "Context: {CONTEXT}\nQuestion: Given the above context, list ten alternative words for "WORD" that are easier to understand.\n"} |
| GPT-3.5-turbo Prompt Without Context |
| {"role": "system", "content": "You are a helpful assistant."}
{"role": "user", "content": "Question: Find ten easier words for "compulsory".\n"}
{"role": "assistant", "content": "1. mandatory\n2. required\n3. essential\n4. forced\n5. important\n6. necessary\n7. obligatory\n8. unavoidable\n9. binding\n10. prescribed\n\n"}
{"role": "user", "content": "Question: Find ten easier words for "WORD"".\n} |

Table 8: The prompt template for one-shot GPT-3.5-turbo in English with and without context. WORD and CONTEXT denote the target word $x$ and context $c_x$, respectively. We modified the template used in Aumiller and Gertz (2022) for the purpose of using *gpt-3.5-turbo* instead of *text-davinici-002*.