# OpenReview forum: "Unsupervised Lexical Simplification with Context Augmentation"
_EMNLP/2023/Conference — EMNLP 2023 Findings_

### Official Review · Reviewer_LjvW · 2023-08-02

**Soundness:** 4

**Excitement:**

3: Ambivalent: It has merits (e.g., it reports state-of-the-art results, the idea is nice), but there are key weaknesses (e.g., it describes incremental work), and it can significantly benefit from another round of revision. However, I won't object to accepting it if my co-reviewers champion it.

**Paper Topic And Main Contributions:**

This paper proposes a way to do lexical simplification by augmenting the target context with other sentences containing the target word in  monolingual data. In addition to modifying a previous approach that samples substitutes given context, they sample other sentences containing the word and cluster them, and within each cluster generate replacements for the target word using the mask token and a pretrained LM to generate substitutes, the most frequent of which are extracted from each cluster. They take the union over these two sets of candidates and rerank with a suite of 4 metrics. On the evaluated datasets their method outperforms the previous work, but it looks like the GPT3.5 models are best. They show even better performance by combining with GPT.

**Reasons To Accept:**

- well-motivated approach to deal with simplification when the word in question is rare (and therefore will get segmented into subword tokens even further)
- good to see that they evaluated against LLM benchmarks for this task, which would presumably be very strong

**Reasons To Reject:**

- incremental improvement over such benchmarks, especially given the complexity and the amount of steps needed. Method is also somewhat incremental on top of Wada et al., 2022

**Reproducibility:**

3: Could reproduce the results with some difficulty. The settings of parameters are underspecified or subjectively determined; the training/evaluation data are not widely available.

**Reviewer Confidence:**

3: Pretty sure, but there's a chance I missed something. Although I have a good feel for this area in general, I did not carefully check the paper's details, e.g., the math, experimental design, or novelty.

---

> ### Author Rebuttal · Authors · 2023-08-29
>
> > incremental improvement over such benchmarks, especially given the complexity and the amount of steps needed.
>
> Our model achieves substantial improvements over the previous unsupervised methods on both lexical substitution and simplification tasks across all languages involved. For instance, Table 1 shows that our method outperforms the previous best unsupervised systems (which differ for each language) by nearly or more than 10% in Accuracy@1 (80.2 vs 65.7 in English, 61.2 vs. 51.9 in Portuguese, and 47.6 vs. 37.0 in Spanish).
>
> When compared against GPT-3.5-Turbo, our model still lags behind it but that’s because (1) our model is fully unsupervised (i.e. using monolingual data only) while GPT-3.5 is supervised on various tasks with human feedback; and (2) our method employs language models that contain far less parameters (e.g. 435M and 800M parameters for DeBERTa-V3 and T5, resp.) than GPT-3.5, which contains 175B parameters. Despite such substantial differences, our model performs very well in English (80.2 and 94.1 in Accuracy@1 and Potential@3) in comparison to the zero-shot performance of GPT-3.5 (82.8 and 92.8). We contend that incremental improvement would be an understatement, given the difference in terms of the amount of resources (both compute and data) to build these models.
>
>
> > Method is also somewhat incremental on top of Wada et al., 2022
>
> While we incorporate Wada et al., 2022 as part of our model, we also identified their critical limitation with regard to retrieving synonyms of rare words: they often retrieve erroneous substitutes when the target word is poorly tokenised into subwords. We explicitly addressed this problem by using FastText and proposing a new context augmentation method, and as a consequence, our model substantially outperforms Wada et al. (e.g. +15% in Accuracy@1 in English as shown in Table 1). On top of that, we also provide a qualitative analysis that shows some potential limitations of GPT-3.5 on the lexical substitution task in the Appendix B. Therefore, we believe that our paper has important scientific contributions, especially for a short paper.

---

### Official Review · Reviewer_iLSG · 2023-08-04

**Soundness:** 3

**Excitement:**

4: Strong: This paper deepens the understanding of some phenomenon or lowers the barriers to an existing research direction.

**Paper Topic And Main Contributions:**

This paper proposes an unsupervised approach to lexical simplification. The authors combine and extend several previously proposed ideas into a system that works reasonably well. Once combined with GPT-3.5, the results become state-of-the-art. It is worth noting that the proposes approach even without GPT-3.5 outperforms previous work but GPT-3.5 on its own is much stronger than the proposed approach (hence the ensemble with GPT-3.5).

**Reasons To Accept:**

- Reasonably clean approach.
- State-of-the-art results.

**Reasons To Reject:**

- Fairly narrow scientific contributions, mostly a well engineered system.

**Reproducibility:**

4: Could mostly reproduce the results, but there may be some variation because of sample variance or minor variations in their interpretation of the protocol or method.

**Reviewer Confidence:**

3: Pretty sure, but there's a chance I missed something. Although I have a good feel for this area in general, I did not carefully check the paper's details, e.g., the math, experimental design, or novelty.

---

> ### Author Rebuttal · Authors · 2023-08-29
>
> > Fairly narrow scientific contributions, mostly a well engineered system.
>
> Beyond the empirical performance of our proposed models, our paper shows that segmenting rare words into subwords does not necessarily lead to better representations, and when the target word is poorly tokenised (e.g. bol ##e), static word embeddings such as FastText can perform better than BERT-sized language models (as shown in Table 1), illustrating a major limitation of the current subword segmentation method (which is used in almost all NLP models these days). Our context argumentation method specifically addresses this problem and achieves strong empirical results, which we argue make important scientific contributions, especially for a short paper (there are also important applications of lexical substitution and simplification as described below). We will clarify our contributions better in the revision.
>
> > Narrow domain/task.
>
> Lexical simplification and substitution tasks are well-established NLP tasks and there has been an extensive line of research that focuses on them. This is evidenced by the fact that they have a long history as shared tasks at NLP workshops, starting from SEMEVAL-2007 (lexical substitution) and SemEval-2012 (lexical simplification), to the most recent one held in the TSAR-2022 workshop on lexical simplification at EMNLP 2022.
>
> These tasks also have some important applications — lexical substitution is applicable to augmenting data on some downstream tasks [1, 2] or embedding watermarks in text for the purpose of tracing the text provenance [3, 4]; and lexical simplification is applicable to creating systems that make text more accessible to people with limited reading skills, such as non-native speakers and people with cognitive disabilities (e.g. dyslexia). These tasks can also probe the ability of recent language models (including GPT-3.5) to understand the meaning of words at a nuanced level. We will provide a better clarification on the implication of these tasks in the revision.
>
> [1] X. Jiao, Y. Yin, L. Shang, X. Jiang, X. Chen, L. Li, F. Wang, and Q. Liu, “TinyBERT: Distilling BERT for natural language understanding,” in Findings of the Association for Computational Linguistics Findings of ACL: EMNLP 2020
>
> [2] Xiang, R., Chersoni, E., Lu, Q., Huang, C.-R., Li, W., & Long, Y. (2021). Lexical data augmentation for sentiment analysis. Journal of the Association for Information Science and Technology, 72(11), 1432–1447.
>
> [3] Xi Yang, Jie Zhang, Kejiang Chen, Weiming Zhang, Zehua Ma, Feng Wang, and Nenghai Yu. 2022. Tracing text provenance via context-aware lexical substitution. In Proceedings of the 36th AAAI Conference on Artificial Intelligence
>
> [4] Jipeng Qiang, Shiyu Zhu, Yun Li, Yi Zhu, Yunhao Yuan, Xindong Wu, Natural language watermarking via paraphraser-based lexical substitution, Artificial Intelligence, Volume 317, 2023,

---

### Official Review · Reviewer_vBu5 · 2023-08-05

**Soundness:** 4

**Excitement:**

4: Strong: This paper deepens the understanding of some phenomenon or lowers the barriers to an existing research direction.

**Paper Topic And Main Contributions:**

The paper proposes an extension over the lexical simplification (and substitution) approach of Wada et al. (2022).
The original approach involves a sampling mechanism and a clustering component based on sentences containing a potential simplifier/substitute.
The methodological extension in this paper is that -- while keeping the previous sampling and clustering step -- it also samples and clusters additional sentences generated by (m)T5  wrt. to the word to be simplified/substituted.
This form of data augmentation is meant to help in cases when the input sequence with the word to be simplified/substituted lacks sufficient context.

The additional use of static embeddings for assessing the aptness of a potential substitute when the input token is segmented into multiple subtokens is motivated by the fact that the transformer based representations of semantically unrelated words that share some common subtokens might be over-represented.
Given this motivation, the use of fasttext is a somewhat unnatural choice as its reliance on the character n-grams that constitute the words also makes it prone to the same issue that the use of static embeddings is meant to mitigate.
The larger vocabulary size of static embeddings and the ease of accessability of fasttext embeddings for all the 3 investigated languages can make the use of fasttext justified, but it would be interesting to see the use of other pre-trained embeddings (e.g. GloVE or word2vec) in place of fasttext (at least for English).

**Questions For The Authors:**

A) What effect does the number of clusters have on the approach?

B) The lexical simplifiers/substitutes originate from two sources, the $M_1$ highest scoring candidate from the original sentence and the $M_2$ candidate with the highest aptness score from the augmented (generated) sentences.  What is the typical overlap between the two sets of candidates?

**Reasons To Accept:**

Good results achieved for lexical simplification/substitution with a simple augmentation-based extension over the approach of (Wada, 2022).

**Reasons To Reject:**

No such weakness have been identified.

**Reproducibility:**

4: Could mostly reproduce the results, but there may be some variation because of sample variance or minor variations in their interpretation of the protocol or method.

**Reviewer Confidence:**

3: Pretty sure, but there's a chance I missed something. Although I have a good feel for this area in general, I did not carefully check the paper's details, e.g., the math, experimental design, or novelty.

---

> ### Author Rebuttal · Authors · 2023-08-29
>
> > the use of fasttext is a somewhat unnatural choice as its reliance on the character n-grams that constitute the words
>
> While FastText employs character n-grams, it also trains a unique representation for each word, which is not shared with any other words (e.g. the word embedding of “her” is constructed by its character n-grams plus the special sequence "<her>”, where “<” and “>” correspond to the beginning and end of token; see Bojanowski et al., 2017 for details). Therefore, FastText embeddings can represent the semantics of rare words considering the contexts they appear and hence mitigate the problem of transformers’ subword representations. We will clarify this in the camera-ready version.
>
> > it would be interesting to see the use of other pre-trained embeddings (e.g. GloVE or word2vec) in place of fasttext (at least for English).
>
> Interesting suggestion. We tried this, using GloVe instead of FastText in English but found similar results; “Wada et al. + Glove” achieves 64.3 and 51.2 in Accuracy@1 and Accuracy@3@top1 while “Wada et al. + FastText” achieves 64.6 and 51.2, respectively. This result suggests that FastText embeddings are not overly affected by the character n-gram information, unlike transformer subword representations.
>
> > What effect does the number of clusters have on the approach?
>
> In our preliminary experiments in English, we tried increasing the cluster size but did not observe much difference. As such, we used the same number of clusters as Wada et al. (i.e. 4 clusters) for simplicity. We did not extensively explore the impact of the cluster size because Wada et al. reported that using 4 clusters achieves almost the same results as using 8 or 16 clusters.
>
> > The lexical simplifiers/substitutes originate from two sources, the M1 highest scoring candidate from the original sentence and the M2 candidate with the highest aptness score from the augmented (generated) sentences. What is the typical overlap between the two sets of candidates?
>
> On the English lexical simplification task, the number of overlap (with M1 = 15 and M2 = 25) is 5.9 on average, but it ranges from 0 to 12 depending on the instances (the variance is 4.7). In Portuguese and Spanish, the average numbers become lower (5.2 and 4.6, respectively) mainly because in those languages more words are segmented into subwords and embedded poorly (as stated in L244-248), which results in more erroneous substitutes generated from Wada et al. (even with FastText). We will include this detail in the revision.

---

### Meta-Review · Area_Chair_MoHZ · 2023-09-19

**Recommendation:** 4

**Metareview:**

The submission proposes data augmentation techniques to improve performance on lexical simplification in an unsupervised way. The reviewers converge towards the view that the current work is exciting; while the proposal is relatively incremental, it is a simple and effective way to improve on previous results.

---

### Decision · Program_Chairs · 2023-10-07

**Decision:**

Accept-Findings

**Comment:**

The submission proposes data augmentation techniques to improve performance on lexical simplification in an unsupervised way. The reviewers converge towards the view that the current work is exciting; while the proposal is relatively incremental, it is a simple and effective way to improve on previous results.